# *Eruca sativa* Meal against Diabetic Neuropathic Pain: An H_2_S-Mediated Effect of Glucoerucin

**DOI:** 10.3390/molecules24163006

**Published:** 2019-08-19

**Authors:** Elena Lucarini, Eleonora Pagnotta, Laura Micheli, Carmen Parisio, Lara Testai, Alma Martelli, Vincenzo Calderone, Roberto Matteo, Luca Lazzeri, Lorenzo Di Cesare Mannelli, Carla Ghelardini

**Affiliations:** 1Department of Neuroscience, Psychology, Drug Research and Child Health—NEUROFARBA—Pharmacology and Toxicology Section, University of Florence, 50139 Florence, Italy; 2CREA-Council for Agricultural Research and Economics, Research Centre for Cereal and Industrial Crops, 40128 Bologna, Italy; 3Department of Pharmacy, University of Pisa, 56126 Pisa, Italy; 4Interdepartmental Research Centre “Nutraceuticals and Food for Health (NUTRAFOOD)”, University of Pisa, 56126 Pisa, Italy; 5Interdepartmental Research Centre of Ageing Biology and Pathology, University of Pisa, 56126 Pisa, Italy

**Keywords:** diabetic neuropathy, neuropathic pain, glucosinolates, *Eruca sativa*, glucoerucin, H_2_S, Kv7 potassium channels

## Abstract

The management of pain in patients affected by diabetic neuropathy still represents an unmet therapeutic need. Recent data highlighted the pain-relieving efficacy of glucosinolates deriving from Brassicaceae. The purpose of this study was to evaluate the anti-hyperalgesic efficacy of *Eruca sativa* defatted seed meal, along with its main glucosinolate, glucoerucin (GER), on diabetic neuropathic pain induced in mice by streptozotocin (STZ). The mechanism of action was also investigated. Hypersensitivity was assessed by paw pressure and cold plate tests after the acute administration of the compounds. Once bio-activated by myrosinase, both *E. sativa* defatted meal (1 g kg^−1^ p.o.) and GER (100 µmol kg^−1^ p.o., equimolar to meal content) showed a dose-dependent pain-relieving effect in STZ-diabetic mice, but the meal was more effective than the glucosinolate. The co-administration with H_2_S scavengers abolished the pain relief mediated by both *E. sativa* meal and GER. Their effect was also prevented by selectively blocking Kv7 potassium channels. Repeated treatments with *E. sativa* meal did not induce tolerance to the anti-hypersensitive effect. In conclusion, *E. sativa* meal can be suggested as a new nutraceutical tool for pain relief in patients with diabetic neuropathy.

## 1. Introduction

The development of neuropathy is a common long-term complication of uncontrolled hyperglycemia and the relief of neuropathic pain still represents a therapeutic challenge in patients affected by diabetes [1]. The management of diabetic neuropathic pain consists basically in improving glycaemic control as a prophylactic therapy and using medications to alleviate pain. Unfortunately, their use is limited by side effects and by the development of tolerance [2,3]. Recent evidence highlighted the beneficial effect of synthetic and naturally occurring H_2_S donors in different types of persistent pain [4,5]. Among the natural compounds able to release H_2_S there are isothiocyanates, which derive from glucosinolate (GSL) hydrolysis mediated by the enzyme myrosinase (β-thioglucoside glucohydrolase, thioglucosidase, EC3.2.1.147) or by the intestinal microflora [6,7,8]. These phytochemicals are contained in almost all the plants belonging to the Brassicaceae family and are responsible for many of their beneficial effects in animals, as well as in humans [6,8,9,10]. The GSL glucoraphanin (GRA) and the derived isothiocyanate sulforaphane (SFN), are the most widely studied due to their potent anti-inflammatory, antioxidant, anti-cancer, antibiotic, as well as neuroprotective effects [6,11,12,13,14,15,16]. SFN has also been tested in humans, demonstrating an improvement of glucose levels in patients with type 2 diabetes [17]. Nevertheless, its effect on the development of diabetic neuropathy was not studied. Recent findings highlighted the anti-hyperalgesic and protective effects of SFN in an animal model of chemotherapy-induced neuropathy [18] and its capacity to potentiate the antinociceptive effects of opioids in animals with inflammatory pain or diabetic neuropathy [19,20]. As in the case of other H_2_S donors, SNF properties are closely related to the release of H_2_S in vivo and the consequent activation of Kv7 potassium channels [4,7,18,21,22,23].

SFN represents a redox couple with erucin, another isothiocyanate which is derived from the metabolism of glucoerucin (GER), the most abundant GSL in *Eruca sativa* spp. oleifera Mill seeds [24,25,26]. Despite the limited studies on erucin, beneficial properties similar to SFN have recently been reported [27,28,29], as its ability to release H_2_S in vitro and to mediate vasodilatation [30,31]. This also led to assume that the effects showed by SFN in vivo could actually be due to its rapid interconversion to erucin [32]. In 2015 Franco et al. developed a system to obtain a food-safe organic material starting from *E. sativa* (*Eruca sativa* Mill. Sel. NEMAT). This pressure defatted oilseed meal enriched of GSL, suitable to produce bakery products, aimed to realize a functional food [33]. The purpose of this study was to evaluate the antihyperalgesic properties (efficacy and tolerance) of *E. sativa* defatted seed meal (DSM), along with that of its active constituent GER, in a model of diabetic neuropathic pain induced by streptozotocin (STZ) in mice. The involvement of H_2_S release and Kv7 modulation in the pain-relieving activity of *E. sativa* derived products were also investigated.

## 2. Results

### 2.1. Eruca sativa Defatted Seed Meal Characterization

*Eruca sativa* DSM was characterized in its main components: Proteins, % residual oil and its fatty acid profile, GSL content and profile, total free phenolic fraction, and myrosinase activity. Proteins were the main component of *E. sativa* DSM accounting for 36% *w*/*w* of dry matter. The mild oil extraction brought to a 20% residual oil component in DSM, which was characterized for fatty acid profile and resulted in particularly rich erucic, linolenic, oleic, and linoleic acids (37%; 16%; 15%; 12%, respectively). The GSL content accounted for total 138 µmol g^−1^, with 98.6 % GER of the total GSLs, and the remaining GRA. Total free phenolic content was 8.9 ± 0.5 mg of gallic acid equivalents (Ge) g^−1^, according to previous studies [34]. Residual myrosinase activity was 8 ± 2 U, with one enzyme unit (U) corresponding to 1 µmol/g DSM of sinigrin transformed in 1 min. The myrosinase activity was comparable to a previous study [28] and low in comparison to the myrosinase activity of cold extracted *E. sativa* DSM, which was about 24 U [28].

### 2.2. Effect of Eruca sativa Defatted Seed Meal and Glucoerucin on Diabetes-Induced Neuropathic Pain

Figure 1 and Figure 2s how, respectively, the effect of acute oral administration of *E. sativa* DSM (0.1–1 g kg^−1^) and GER (30–100 μmol kg^−1^) in STZ-treated animals with and without myrosinase bioactivation. Four weeks after the STZ injection, mice showed a significantly decreased latency to pain-related behaviours induced by a noxious mechanical stimulus (Paw pressure test, Figure 1a and Figure 2a) as well as by a thermal non-noxious stimulus (cold plate test, Figure 1b and Figure 2b), compared with control mice treated with vehicle (Figure 1 and Figure 2). *Eruca sativa* DSM was able to dose-dependently relieve pain in STZ-treated mice, increasing the paw withdrawal latency to the value of controls 30 min after the administration of both doses 0.3 and 1 g kg^−1^. Furthermore, the higher dose (1 g kg^−1^) of *E. sativa* DSM showed a long-lasting effect, since the animal’s pain threshold was significantly increased up to 90 min after the injection (Figure 1a). The pain-relieving effect of *E. sativa* DSM was also observed in the cold plate test, though it is significant only at the higher dose (Figure 1b). GER, administered in an equivalent dose to that contained in the *E. sativa* DSM (100 μmol kg^−1^) significantly reduced mechanical hyperalgesia in STZ-treated animals, though it was less effective than the meal. By contrast, the 3-fold lower dose of GER (30 μmol kg^−1^) was completely ineffective (Figure 2a). The same trend was observed in the cold plate test (Figure 2b). It is worth noting that both the solutions of *E. sativa,* DSM and GER, lost their effect on pain without the myrosinase-mediated bioactivation (Figure 1 and Figure 2).

Figure 3 shows the effect of the repeated treatment with *E. sativa* DSM on STZ-induced neuropathic pain in mice. The defatted seed meal was administered once daily for 8 consecutive days to evaluate the development of tolerance in these animals. The Figure shows the animal pain threshold 60 min after the treatments. The acute pain-relieving effect of *E. sativa* DSM (1 g kg^−1^) remained constant over time, without the onset of tolerance. On the other hand, the repeated administration of *E. sativa* DSM did not influence the animal basal threshold, which was not different from that of STZ + vehicle-treated animals before the compound administration.

### 2.3. Role of Isothiocyanates and H_2_S-Release in the Anti-Hyperalgesic Effect of Eruca sativa Defatted Seed Meal and Glucoerucin 

To evaluate the role of H_2_S in the anti-hyperalgesic effect showed from *E. sativa* DSM and GER, we administered the compounds in mixture with oxidized glutathione (GSSG), a compound able to bind the isothiocyanates, preventing the release of H_2_S [35,36,37], and hemoglobin (Hb), a molecule able to bind H_2_S [38]. The co-administration with GSSG (20 mg kg^−1^ po) was able to fully prevent the anti-hyperalgesic effect of *E. sativa* DSM (1 g kg^−1^) as well as that of GER (100 μmol kg^−1^): the pain threshold of animals treated with both these substances in mixture with GSSG was not significantly different from that of STZ + vehicle-treated animals (Figure 4a,b). The same result was observed by systemically administering GSSG (20 mg kg^−1^ sc) in concomitance with the oral administration of *E. sativa* DSM and GER (Figure 4a,b). Both the oral and the subcutaneous administration of GSSG (20 mg kg^−1^) in STZ-treated animals did not elicit effects on animal pain threshold (Appendix A). The effect of the tested compounds was also abolished by co-administering Hb (300 mg kg^−1^ po) to both products (Figure 4c,d). 

### 2.4. Involvement of Kv7 Potassium Channels in the Pain-Relieving Effect of Eruca sativa Defatted Seed Meal and Glucoerucin 

To study the involvement of the Kv7 potassium channel in the anti-neuropathic effect of *E. sativa* DSM and GER, the selective Kv7 blocker, XE991, was intraperitoneally administered in concomitance with both the substances. Figure 5 shows the effect induced by *E. sativa* DSM (1g kg^−1^ po) and GER (100 μmol kg^−1^ po) on diabetes-induced neuropathic pain in comparison with that obtained by pre-treating the animals with XE991 (1 mg kg^−1^ ip). The anti-hyperalgesic effect of *E. sativa* DSM and GER was fully prevented by the intraperitoneal administration of XE991, as highlighted in Figure 5a,b, respectively. 

## 3. Discussion

This work, for the first time, highlighted the pain-relieving properties of *E. sativa*. In particular, we demonstrated that *E. sativa* DSM, along with its main GSL, GER, can counteract pain in animals affected by diabetic neuropathy induced by STZ. As in the case of other GSLs [18], the effect was mediated by H_2_S release and the consequent activation of Kv7 potassium channels. However, the effect of the *E. sativa* DSM was not only attributable to the content of GER, suggesting a synergism between the GSLs and the other phytochemicals contained in this nutraceutical product.

Diabetic peripheral neuropathy is a distressing disease of the nerves in the hands and feet, it is a common complication of both type 1 and type 2 diabetes [1,39]. The symptoms most commonly experienced in these patients are burning, electric-shock type and sharp pains [39,40,41], while aching, itching, and cold pain are common but less prevalent manifestations [41,42]. These symptoms increase at night, predictably interfering with sleep [43]. In pre-clinical studies, the STZ model of diabetes has been shown to be associated with sensory changes including allodynia and hyperalgesia which develop starting in the few weeks following STZ administration [44,45].

The management of diabetic neuropathy involves maintaining good glycaemic control by drugs as well as by lifestyle modification, like diet and exercise [46,47,48]. Current therapies aiming to relieve neuropathy-related symptoms, such as pain, are helpful only in one-third of patients with 50% of efficacy [27], often achieved with troublesome side effects and low levels of satisfaction [47,49].

Evidence collected in the last few years supports a diet-based approach in the management of different types of diseases, including chronic pain syndromes [50,51,52]. Among the most largely studied nutraceutical products there are the GSLs, secondary metabolites that can be found in different concentrations among cruciferous vegetables [6,18]. It is thought that GSLs, together with myrosinase, are a part of a defense mechanism implemented by plants to protect themselves against biotic and abiotic stress [53]. Recently we discovered that GSLs along with their hydrolysis product, isothiocyanates, are strongly effective in relieving neuropathic pain induced by chemotherapy [18]. Interestingly, other natural and synthetic isothiocyanates-based compounds also showed broadly pain-relieving properties in animal models of persistent pain [4,5].

*Eruca sativa* is an edible plant indigenous of the Mediterranean area that has been traditionally used for its medicinal properties. Similar to other Brassicaceae, the characteristic pungent taste and odor of *E. sativa* leaves were attributed to GSLs. This plant has a high vitamin C content and is known for various health-promoting effects, including improvement of blood circulation, diuretic, and anti-inflammatory properties [27,54,55,56]. *Eruca sativa* DSM represents an enriched functional food formulated to modulate the release over time of GSLs degradation products, in order to obtain the most effective biological activity [28,57]. Accordingly, we found that the employment of *E. sativa* DSM led to a more strong and long-lasting effect in comparison with GER. The effect of both the compounds is dependent on the bioactivation mediated by myrosinase, indicating as being mainly responsible for pain relief the isothiocyanates derived from the hydrolysis of GER, erucin. The obtained result opened the way to two different hypotheses. The first is that the effect of *E. sativa* DSM is mediated not only by GSLs and the derived isothiocyanates but also by other constituents, as phenols and fatty acids. Indeed, the beneficial effects of antioxidant compounds, such as phenols, against neuropathic pain are well known [58,59,60,61,62]. On the other hand, the effect of *E. sativa* DSM is likely attributable to the GSL content [4,18], since its effect was completely abolished by co-administering the H_2_S scavenger as well as by the oxidative cleavage of the isothiocyanate mediate by the disulfide GSSG. In particular, GSSG, systemically administered, was able to prevent the pain-relieving effect, indicating that the isothiocyanate erucin, once adsorbed in the gut, reaches the bloodstream where it probably releases H_2_S. On the other hand, the administration of GSSG in a mixture with the bioactivated compounds could prevent its intestinal absorption.

The effect of *E. sativa* was also prevented blocking Kv7 potassium channels, confirming again the direct involvement of GSLs in the pain-relieving effect [4,18].

This evidence, which seems to diminish the importance of the other constituents of the meal own effect on pain modulation, actually move the attention to the second hypothesis, namely that this formulation could be able to modulate the release of the GSLs, improving their pharmacological profile. In fact, it is known that the efficacy of H_2_S donors is closely linked to their pharmacokinetics—slower is the release of H_2_S, higher is the efficacy [4,5]. This is mainly due to the bell-shape beneficial effect showed by H_2_S in the organism [63]. In this context, the peculiar formulation of *E. sativa* DSM could determine a slow release of GSLs, allowing a sustained availability of isothiocyanates and consequently H_2_S in vivo, and avoiding quick exhaustion of the pain-relieving effect.

A peculiar feature of GER is that it needs to be preventively bioactivated by myrosinase in solution before the administration in vivo. The same was observed with the *E. sativa* DSM, strengthening once again the importance of its glucosinolates content in the pain-relieving effect. This chemical property makes GER different from other GLSs, such as GRA. In fact, in our previous work, we tested the anti-hyperalgesic effect of this GLS and the derived isothiocyanate, SFN, in a model of neuropathic pain induced by chemotherapy [18]. In that case, we did not have to pre-treat the GLS with the enzyme myrosinase to get the effect. By mixing myrosinase with GER or *Eruca sativa* DSM, we are delivering the isothiocyanate erucin in vivo. It likely means that GER is ineffective. In the absence of myrosinase, GLS can be hydrolyzed to isothiocyanates by the bacteria that constitute the gut microflora. Nevertheless, it was found that the microbiome only supports poor hydrolysis, unless exposed to dietary GLSs for a period of days [64]. On the other hand, the microbiota could not have the time necessary to hydrolyze GER in the lumen if it is quickly absorbed or metabolized in different products in the gut. Anyway, the fact that these compounds need to be bioactivated does not preclude the possibility of using them but it is a point to take into account for the treatment protocol.

Another important point to consider is that the effect of *E. sativa* is eligible only after the acute administration and it does not show a therapeutic effect on the neuropathy, which also persists after the repeated treatment. By contrast, the analog, GRA, and the derived isothiocyanate, sSFN, were able to prevent the development of the neuropathy induced by chemotherapy [18]. This observation suggests that the administration of the GSLs can counteract the pathophysiological mechanisms that lead to the instauration of the neuropathies, such as the oxidative stress [5,65,66], but cannot revert it after its establishment. Anyway, in the case of diabetic neuropathy, it is difficult to apply a preventive treatment at clinical level since this type of disease is a long-term complication of uncontrolled hyperglycemia [2,39,40]. In fact, a proper treatment of hyperglycemia is the most effective and straight forward method to prevent neuropathy in diabetic patients. So, another interesting possibility could be to employ *E. sativa*-derived GLSs as a nutraceutical approach to manage pain in diabetes patients who have already developed the neuropathy, since it still represents a therapeutic problem [48,49]. Indeed, although *E. sativa* does not appear to be a resolving treatment for diabetic neuropathy, its acute anti-hyperalgesic efficacy joined to the characteristics of food supplements suggest its clinical use for treating neuropathic pain as monotherapy or in combination with drugs for sparing dosages and enhance efficacy.

In this perspective, *E. sativa* DSM proved to be a valid innovative formulation to guarantee proper delivery of GSLs in vivo and to enhance their therapeutic efficacy against pain, laying the bases for its rational use in patients affected by diabetic neuropathy.

## 4. Materials and Methods

### 4.1. Eruca sativa Defatted Seed Meal Production and Characterization

*Eruca sativa* Mill. Var. NEMAT was grown during the season 2014–2015, within a plot with a size of 1100 m^2^, adopting a minimum agronomical input approach [23]. The cultivation was carried out at CREA experimental farm located at Budrio (Bologna) in the Po Valley area (Emilia Romagna region, 44°32′00” N; 11°29′33” E, altitude 28 m a.s.l.). The area was characterized by flat land with alluvial deep loamy soil with medium level content of total nitrogen and organic matter content. After harvesting, *E. sativa* seeds were accurately threshed with fixed small-scale threshing equipment and air-dried to reduce the high residual moisture content. *Eruca sativa* seeds were defatted using a small continuous seed crusher machine (Bracco Company model Elle.Gi type 0.90) at a temperature-controlled procedure, during which temperature was maintained at a maximum of 70 °C. *Eruca sativa* DSM was characterized by moisture, proteins, residual oil, GSL, total free phenolic content, and residual myrosinase activity according to the following methods:(1)Moisture content was determined to evaluate the difference between its weight before and after oven-drying at 105 °C for 12 h.(2)Proteins were determined from the total content of nitrogen determined using the elemental analyzer LECO CHN TruSpec according to the American Society for Testing Materials (ASTM D5373).(3)Residual oil content was determined by the standard Soxhlet extraction method using hexane as a solvent and characterized for its fatty acid composition by the UNI EN ISO 5508 method (1998) [67]. Fatty acid composition of residual oil was analyzed after trans-methylation in 2N KOH methanol solution. Fatty acid methyl esters (FAMEs) were evaluated by gas chromatography and the internal normalization method [68] was used for determining the fatty acid profile.(4)Glucosinolate content was determined following the ISO 9167-1 method with some minor modifications [69]. Briefly, 250 mg DSM were extracted in 70% ethanol at 80 °C. One milliliter of crude extract was loaded onto a DEAE Sephadex A-25 (GE Healthcare, Freiburg, Germany) mini-column. After washing with 25 mM acetate buffer (pH 5.6), GSLs were desulfated by adding purified sulfatase (200 µL, 0.35 U/mL). The desulfo-GSLs were eluted in water (HPLC grade) and detected in HPLC-UV [69] monitoring their absorbance at 229 nm. They were identified with respect to their UV spectra and retention time, according to our library [70], and their amounts were estimated using sinigrin as an internal standard. Each extraction and analysis was performed in triplicate.(5)Total free phenolic content was assayed with the Folin–Ciocalteu method according to [71]. Values are the mean ± SD of three independent extractions by four replicates for each measurement. *Eruca sativa* DSM extracts were obtained in acidified ethanol, ethanol/1 N HCl (85:15; *v*/*v*), after 30 min at 21 °C in 40 kHz ultrasonic bath (Sonica Sweep System, Soltec). The supernatants of a triple extraction procedure were collected and maintained at −20 °C in the dark for 48 h to facilitate macromolecule precipitation. Five serial dilutions of the filtered extracts were assayed at 765 nm, 20 °C, in an Infinite M200 NanoQuant Plate reader (Tecan, Switzerland). The slope of each calibration curve was compared to a standard gallic acid calibration curve (range 0.3–27 µg ml^−1^, r^2^ = 0.9972). The slope ratio of sample/standard curves was calculated, and results were expressed as mg of GAE per g of DSM.(6)Myrosinase activity was determined by the pH-stat technique according to [28]. Briefly, 300 mg of *E. sativa* DSM were loaded in 15 mL of 1% NaCl into a reaction cell at 37 °C in a DL50 pH-Stat titrator (Mettler Toledo, Switzerland). The reaction started by adding 0.5 mL of 0.5 M sinigrin solution in distilled water, after 8–10 min of conditioning, and was monitored following NaOH additions used to maintain pH constant at 6.5, versus time in minutes. The assay was carried out in triplicate. One enzyme unit (U) corresponded to 1 μmol/g DSM of sinigrin transformed in 1 min.

### 4.2. Isolation of Glucoerucin

Glucoerucin was isolated starting from *E. sativa* seeds, as previously described [28]. Briefly, seeds of *E. sativa* from the Brassicaceae collection at CREA-CI, Bologna, Italy [36] were crushed in boiling 70% ethanol and the GSL was isolated as K^+^ salt by two sequential steps, including ion exchange and size exclusion chromatography. Isolated GER preparation was analyzed by HPLC-UV after enzymatic desulfation according to [69] and it was identified by UV spectra and HPLC retention times according to our library [70]. The purity of GER was 97% as indicated by HPLC-UV chromatograms and 91 ± 2% on weight basis estimated using sinigrin as an internal standard. It was stored until use at −20 °C. Erucin was produced in-situ through myrosinase-catalyzed hydrolysis, as previously described [30]. Myrosinase (32 U/mL), was isolated from the seeds of *Sinapis alba* L. according to [72]. One unit of myrosinase activity is defined as the amount of enzyme capable of hydrolyzing 1 µmol sinigrin, per min, at pH 6.5 and 37 °C

### 4.3. Animals

Animals were male C57BL/6 mice (Envigo, Varese, Italy) weighing approximately 22–25 g at the beginning of the experimental procedure were used. Animals were housed in CeSAL (Centro Stabulazione Animali da Laboratorio, University of Florence) and used at least 1 week after their arrival. Ten mice were housed per cage (size = 26 × 41 cm); animals were fed with a standard laboratory diet and tap water ad libitum and kept at 23 ± 1 °C with a 12-hr light/dark cycle, with light at 7 a.m. All animal manipulations were carried out according to the Directive 2010/63/EU of the European Parliament and of the European Union council (September 22, 2010) on the protection of animals used for scientific purposes. The ethical policy of the University of Florence complies with the Guide for the Care and Use of Laboratory Animals of the U.S. National Institutes of Health (NIH Publication No. 85–23, revised 1996; University of Florence assurance number: A5278-01). Formal approval to conduct the described experiments was obtained from the Animal Subjects Review Board of the University of Florence. Experiments involving animals have been reported according to ARRIVE guidelines [39]. All efforts were made to minimize animal suffering and to reduce the number of animals used.

### 4.4. Induction of Diabetic Neuropathy in Mice

Mice were intraperitoneally administered with STZ, (Sigma Aldrich, Milan, Italy) 100 mg kg^−1^, followed three days after with a second dose of STZ 50 mg kg^−1^ [73]. Since streptozotocin has stability problems [74], the solution was prepared immediately before the injection. To maintain cleanliness and avoid the development of any infection due to excessive urination, animal bedding was changed frequently. Pain threshold was investigated each week after the injection of STZ and tests were performed once neuropathy was established in mice.

### 4.5. Assessment of Mechanical Hyperalgesia

Mechanical hyperalgesia was determined by measuring the latency in seconds to withdraw the paw away from a constant mechanical pressure exerted onto the dorsal surface [75]. A 15 g calibrated glass cylindrical rod (diameter = 10 mm) chamfered to a conical point (diameter = 3 mm) was used to exert the mechanical force. The weight was suspended vertically between two rings attached to a stand and was free to move vertically. A single measure was made per animal. A cut off time of 40 s was used.

### 4.6. Assessment of Thermal Allodynia

The animals were placed in a stainless-steel box (12 × 20 × 10 cm) with a cold plate as floor. The temperature of the cold plate was kept constant at 4 °C ± 1 °C. Pain-related behavior (licking of the hind paw) was observed, and the time (seconds) of the first sign was recorded. The cut-off time of the latency of paw lifting or licking was set at 30 s [76]. The results were expressed by the licking latency resulting from the compounds acute administration.

### 4.7. Compounds Administration

*Eruca sativa* DSM, GER, and myrosinase were produced according to the method described above. Compounds were acutely administered as follows. The doses of *E. sativa* DSM (0.1–1 g kg^−1^ po) were chosen based on previously published H_2_S releasing and antinociceptive properties of synthetic and natural ITCs [4,5]. The doses of GER (30–100 μmol kg^−1^ po) used are equivalent to those contained in *E. sativa* DSM (0.3–1 g kg^−1^ po). Both *E. sativa* DSM and GER were bioactivated by adding 30 μL mL^−1^ of myrosinase (32 U mL^−1^) 15 min before administering them in the animals. Behavioral tests were carried out 30, 60, 90, and 120 min after the injection. Afterward, repeated oral administrations of *E. sativa* DSM (0.3–1 g kg^−1^) were carried out daily in the mice after the establishment of the diabetic neuropathy. Behavioral tests were performed once daily after the acute administration. In additional experiments, *E. sativa* DSM (1g kg^−1^) and GER (100 μmol kg^−1^) were administered in mixture with human hemoglobin 4.6 μmol kg^−1^ (300 mg kg^−1^; Hb; Sigma-Aldrich, Italy) and with glutathione 65 μmol kg^−1^ (20 mg kg^−1^; GSSG; Sigma-Aldrich, Milan, Italy). The effect of the subcutaneous administration of GSSG 15 min before *E. sativa* DSM (1g kg^−1^ po) and GER (100 µmol kg^−1^ po) was also evaluated. The Kv7 potassium channel blocker XE991 (Tocris Bioscience, Italy; 2.66 μmol kg^−1^; 1 mg kg^−1^; [77]) was dissolved in saline solution and intraperitoneally administered in concomitance with the tested compounds injection.

### 4.8. Statistical Analysis

Behavioral measurements were performed on 10 mice for each treatment carried out in two different experimental sets. Investigators were blind to all experimental procedures. Results were expressed as mean ± S.E.M. The analysis of variance of data was performed by one-way analysis of variance, and a Bonferroni’s significant difference procedure was used as post hoc comparison. *p* values of less than 0.05 or 0.01 were considered significant. Data were analyzed using the “Origin 9” software (OriginLab, Northampton, MA, USA).

## Figures and Tables

**Figure 1 molecules-24-03006-f001:**
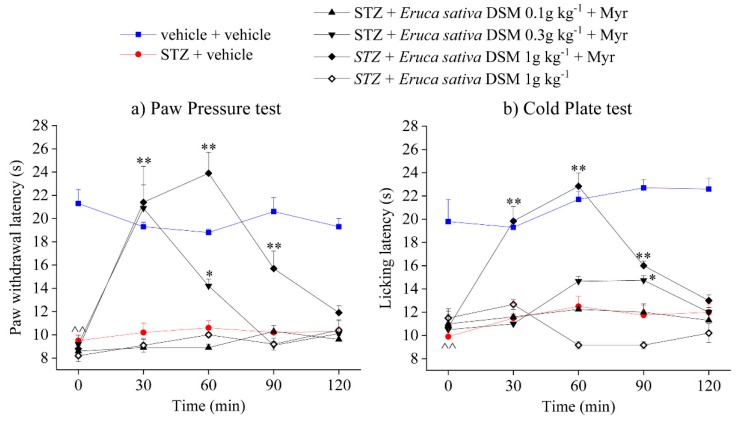
Effect of acute administration of bioactivated *Eruca sativa* defatted seed meal on streptozotocin (STZ)-induced neuropathic pain. The response to both a mechanical and a thermal stimulus was evaluated by measuring the latency (s) to pain-related behaviors; (**a**) withdrawal or (**b**) licking of the paw. *Eruca sativa* defatted seed meal (DSM) (0.1–1 g kg^−1^) was bioactivated by adding 30 μL mL^−1^ of myrosinase (Myr) (32 U mL^−1^) 15 min before the oral administration in STZ-treated animals. Tests were performed 30, 60, 90, and 120 min after the injection. ^^ *p* < 0.01 versus vehicle + vehicle-treated mice; * *p* < 0.05 and ** *p* < 0.01 versus STZ + vehicle-treated mice.

**Figure 2 molecules-24-03006-f002:**
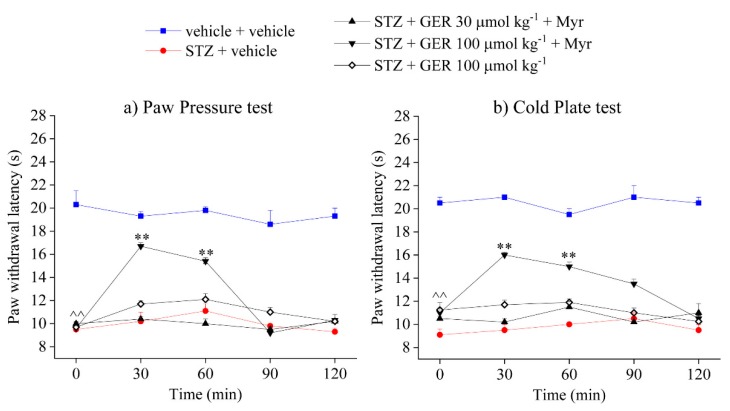
Effect of acute administration of bioactivated glucoerucin (GER) on streptozotocin (STZ)-induced neuropathic pain. The response to both a mechanical and a thermal stimulus was evaluated by measuring the latency (s) to pain-related behaviors; (**a**) withdrawal or (**b**) licking of the paw. GER (30–100 µkg^−1^) was bioactivated by adding 30 μL mL^−1^ of myrosinase (32 U mL^−1^) 15 min before the oral administration in STZ-treated animals. Tests were performed 30, 60, 90, and 120 min after the injection. ^^ *p* < 0.01 versus vehicle + vehicle-treated mice; * *p* < 0.05 and ** *p* < 0.01 versus STZ + vehicle-treated mice.

**Figure 3 molecules-24-03006-f003:**
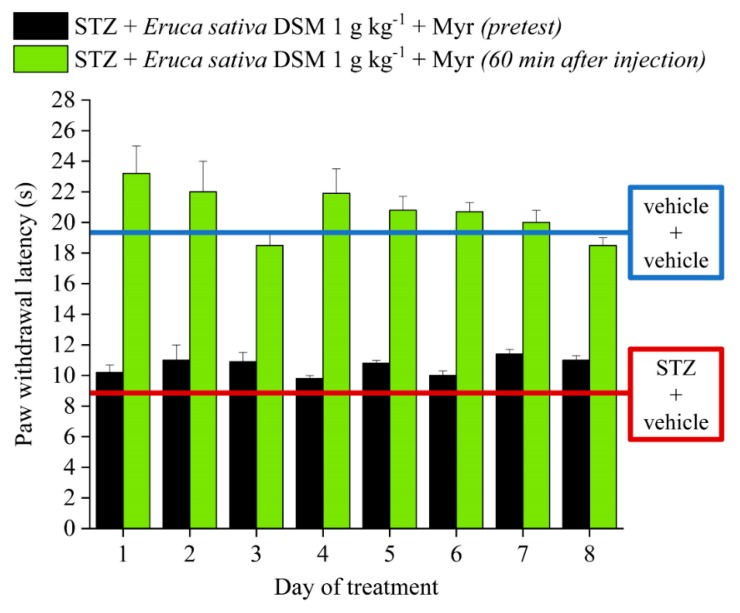
Effect of the repeated treatment with *Eruca sativa* defatted seed meal (DSM) on streptozotocin (STZ)-induced neuropathic pain. The response to a mechanical stimulus was evaluated by measuring the latency (s) to pain-related behaviors (paw withdrawal). The myrosinase (Myr)-bioactivated *Eruca sativa* DSM (1 g kg^−1^) was administered once daily for 8 consecutive days in STZ-treated animals (once neuropathy was established) and pain threshold was assessed before and 60 min after the injection. ^^ *p* < 0.01 versus vehicle + vehicle-treated mice; * *p* < 0.05 and ** *p* < 0.01 versus STZ + vehicle-treated mice.

**Figure 4 molecules-24-03006-f004:**
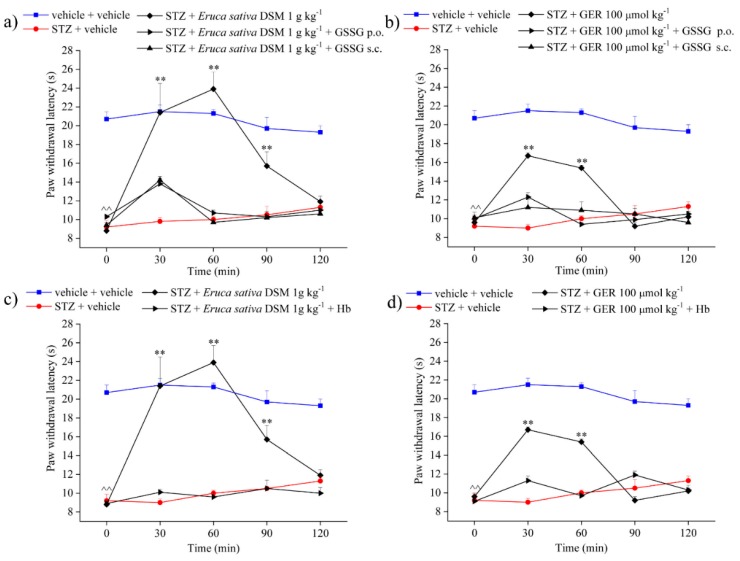
Role of H_2_S in the pain-relieving effect of *Eruca sativa* defatted seed meal (DSM) and glucoerucin (GER). The response to a mechanical stimulus was evaluated by measuring the latency(s) to pain-related behavior (paw withdrawal). Oxidized glutathione (GSSG) (20 mg kg^−1^) was orally and subcutaneously administered in concomitance with both (**a**) Myr-bioactivated *Eruca sativa* DSM (1 g kg^−1^) and (**b**) GER (100 µmol kg^−1^); tests were performed 30, 60, 90, and 120 min after injection. (**c**) Myrosinase (Myr)-bioactivated *Eruca sativa* DSM (1 g kg^−1^) and (**d**) GER (100 µmol kg^−1^) were orally administered alone or in mixture with human hemoglobin (Hb) (300 mg kg^−1^); tests were performed 30, 60, 90, and 120 min after injection. ^^ *p* < 0.01 versus vehicle + vehicle-treated mice; * *p* < 0.05 and ** *p* < 0.01 versus streptozotocin (STZ) + vehicle-treated mice.

**Figure 5 molecules-24-03006-f005:**
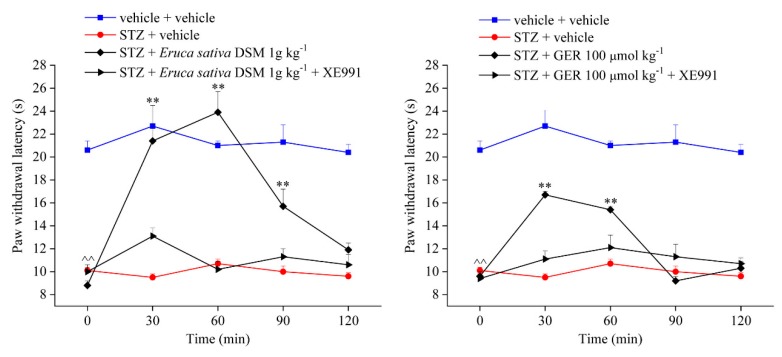
Involvement of Kv7 potassium channels in the pain-relieving effect of *Eruca sativa* defatted seed meal (DSM) and glucoerucin (GER). The response to a mechanical stimulus was evaluated by measuring the latency to pain-related behaviors (paw withdrawal). (**a**) Myrosinase (Myr)-bioactivated *Eruca sativa* DSM (1 g kg^−1^) and (**b**) GER (100 µmol kg^−1^) were orally administered in concomitance with XE991 (1 mg kg^−1^ ip); tests were performed 30, 60, 90, and 120 min after injection. ^^ *p* < 0.01 versus vehicle + vehicle-treated mice; * *p* < 0.05 and ** *p* < 0.01 versus streptozotocin (STZ) + vehicle-treated mice.

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
