# Peer review of "Eruca sativa Meal against Diabetic Neuropathic Pain: An H2S-Mediated Effect of Glucoerucin"

_molecules, 2019, doi:10.3390/molecules24163006_

Round 1

Reviewer 1 Report

The authors has conducted a pioneering systematic research regarding discovery, biological evaluation and mechanism study for the potential natural-recurring therapeutic reagent with antihyperalgesic effect. The method and experimental design has been taken full of consideration. I think their study will provide solid robust reference for the future study in this plant especially for the potential unknown promising bioactive gluosinolate or other class of compounds contributing to effect of DSM. Several minor suggestions for the authors:

I suggest to remove Figure 1 as ll the information included in the Table can be presented very straightforwardly in the paragraph

I am curious about have the authors also explored the effect of repeated administration with GER. Although GER at 100 umol kg-1 has shown a weaker activity compared with DSM, it is still worthwhile to see whether its effect can be improved by repeated administrations.

Reviewer 2 Report

The manuscript (id molecules­537715), “Eruca sativa meal against diabetic neuropathic pain: a H2S­mediated effect of glucoerucin,” reported the effect of Eruca sativa defatted seed meal on diabetic neuropathic pain using mice model. The manuscript is not well prepared. The work is interesting. However, this reviewer believes that the current version of the manuscript is not acceptable in Molecules.

Specific comments

1. Page 2, Figure 1 is not a figure, and should be Table 1.

2. Page 2, Figure 1: Oil should be more specific in chemical components or names, for example fatty acids.

3. Page 2, Figure 1: GSL and TPC should be spelled out.

4. Page 3, Figure 2: “myrosinase (Myr)” should be used in figure legend.

5. Page 5, Figure 5: GER, GSSG, Hb should be spelled out.

6. Page 6, Figure 6: XE991 should be explained.

7. Page 8: The authors should provide more details in terms of methodology for moisture content measurement.

8. Page 8: The authors should provide more details for fatty acid analysis, what instrumentation and experimental conditions?

9. Page 8: More details on glucosinolate content analysis is needed.

10. Page 8: More details on phenolic content analysis is needed, for example, instrumentation and experimental conditions.

11. Page 8: More details on myrosinase activity determination is required, instrumentation, method, and experimental conditions.

12. Conclusion is somewhat not well summarized and hard to read. The authors should highlight the improvement or innovation of this work over the previous reports in literature.

13. A working model of the proposed treatment against diabetic neuropathic pain should be provided to readers.

Reviewer 3 Report

This is an extremely interesting and a very important paper.  It should be published in a good journal (e.g. Molecules).  However, the English grammar must be improved markedly, and the paper suffers from lack of clarity over and above the grammar.  Most perplexing is the conflation of terminology and lack of clarity about what’s happening.  If you mix myrosinase and GER, you are delivering the isothiocyanate erucin.  If not, you are delivering GER (glucoerucin).  Please, when you re-write, make sure to be absolutely clear to the reader what you’re talking about.  Furthermore, you neglect the myrosinase to be found, and which will be active, in the animal’s gut.  Certainly when you do injection vs. oral delivery, you bypass that route, but nothing is ever really said about that.  The results are all there, I feel, but the story is not being told well enough to let us walk through the experimental adventure properly with you, and to understand where you wind up.

A few additional points: 

It wasn’t abundantly clear that the evidence showed that isothiocyanates were not BINDING to the hemoglobin, rather than hemoglobin ablating H2S production.  The experiments in Figs 2 and 3 did not show that.

Page 6, end of 1st par. In section 3: you talk about “the nutraceutical product”.  What product??

On page 7 you say “A difference between GRA [18] and GER lies in the fact that the second one needs to be preventively bioactivated by myrosinase in solution before the administration in vivo.”  None of this is abundantly clear from the evidence you present.

Importantly, in Section 2.1, you say “Residual myrosinase activity was low in comparison to the myrosinase activity of E. sativa seeds, which is about 24 U”.  but what IS it.  Low. . .  how low?  Please tell us.  And when you say 24 U (for the seeds), is that Units per ton, per tonne, per seed, per gram, per femtogram?

Round 2

Reviewer 2 Report

The revised manuscript is acceptable for publication.